**Data Availability Statement:** All relevant data are within the paper and its Supporting Information files.

# The correlation of salivary telomere length and single nucleotide polymorphisms of the *ADIPOQ*, *SIRT1* and *FOXO3A* genes with lifestyle-related diseases in a Japanese population

**Xiao Han**[1], **Ryo Kubota**[2], **Ken-ichi Tanaka**[1], **Hiroyuki Hayashi**[1], **Miyuki Seki**[3], **Nobue Sakai**[2], **Noriko Kawaguchi-Ihara**[2], **Kyoko Arakawa**[2], **Ikuo Murohashi**[1,4]*

1 Center for University-Wide Education, School of Health and Social Services, Saitama Prefectural University, Koshigaya, Saitama, Japan, 2 Department of Health Sciences, School of Health and Social Services, Saitama Prefectural University, Koshigaya, Saitama, Japan, 3 Department of Nursing, School of Health and Social Services, Saitama Prefectural University, Koshigaya, Saitama, Japan, 4 Internal Medicine, Seibu-Iruma Hospital, Iruma, Saitama, Japan

* murohashi-ikuo@spu.ac.jp

## Abstract

### Background

It has been reported that genetic factors are associated with risk factors and onset of lifestyle-related diseases, but this finding is still the subject of much debate.

### Objective

The aim of the present study was to investigate the correlation of genetic factors, including salivary telomere length and three single nucleotide polymorphisms (SNPs) that may influence lifestyle-related diseases, with lifestyle-related diseases themselves.

### Methods

In one year at a single facility, relative telomere length and SNPs were determined by using monochrome multiplex quantitative polymerase chain reaction and TaqMan SNP Genotyping Assays, respectively, and were compared with lifestyle-related diseases in 120 Japanese individuals near our university.

### Results

In men and all participants, age was inversely correlated with relative telomere length with respective *p* values of 0.049 and 0.034. In men, the frequency of hypertension was significantly higher in the short relative telomere length group than in the long group with unadjusted *p* value of 0.039, and the difference in the frequency of hypertension between the two groups was of borderline statistical significance after adjustment for age (*p* = 0.057). Furthermore, in men and all participants, the sum of the number of affected lifestyle-related

**Funding:** This work was supported by 2015 Graduate Student Exchange (XH) and Grant-in-Aid for Encouragement of Scientists Research Funds (IM), Saitama Prefectural University. The funders had no role in study design, data collection and analysis, decision to publish, or preparation of the manuscript.

**Competing interests:** The authors have declared that no competing interests exist.

diseases, including hypertension, was significantly higher in the short relative telomere length group than in the long group, with *p* values of 0.004 and 0.029, respectively. For *ADIPOQ rs1501299*, men's ankle brachial index was higher in the T/T genotype than in the G/G and G/T genotypes, with *p* values of 0.001 and 0.000, respectively. For *SIRT1 rs7895833*, men's body mass index and waist circumference and all participants' brachial-ankle pulse wave velocity were higher in the A/G genotype than in the G/G genotype, with respective *p* values of 0.048, 0.032 and 0.035. For *FOXO3A rs2802292*, women's body temperature and all participants' saturation of peripheral oxygen were lower in the G/T genotype than in the T/T genotype, with respective *p* values of 0.039 and 0.032. However, relative telomere length was not associated with physiological or anthropometric measurements except for height in men (*p* = 0.016). *ADIPOQ rs1501299* in men, but not the other two SNPs, was significantly associated with the sum of the number of affected lifestyle-related diseases (*p* = 0.013), by genotype. For each SNPs, there was no significant difference in the frequency of hypertension or relative telomere length by genotype.

## Conclusion

Relative telomere length and the three types of SNPs determined using saliva have been shown to be differentially associated with onset of and measured risk factors for lifestyle-related diseases consisting mainly of cardiovascular diseases and cancer.

## Introduction

Telomeres are a repetitive nucleotide sequences of TTAGGG at the end of a chromatid, maintaining the stability of chromosomes to avoid deterioration and fusion with other chromosomes [1]. Telomere length (TL) is influenced by genetic factors, with previous studies reporting heritability estimates ranging from 34 to 82% [2, 3]. This parameter is also cumulatively shaped by nongenetic influences throughout human life. In contrast, single nucleotide polymorphisms (SNPs) are inherited from parents and transmit heritable events [4].

Quantitative trait locus studies have mapped putative loci that may be involved in regulating TL to human chromosomes *3q26.1*, *10q26.13* and *12q12.22* [5]. Indeed, a number of recent genome-wide association studies (GWAS) identified common SNPs near the *telomerase RNA component* (*TERC*) associated with TL in European and Chinese populations [5, 6]. The strongest associations with TL were reported for the SNPs *rs12696304* and *rs16847897* near *TERC* on *3q26*. Although their functions are unclear, these genes appear to be involved in the maintenance of chromosome structures [7].

Research on the genetics of human longevity has identified hundreds of genes associated with longevity, but polymorphisms in only two genes (*APOE* and *FOXO3*) have demonstrated strong and consistent replications across multiple, diverse human populations [8]. The correlation between genetic factors such as TL and SNPs and lifestyle-related diseases (LRDs) is also still controversial [8–15]. Furthermore, these studies [5, 6, 8–15] were conducted in different population cohorts, including Chinese, American and European cohorts, and in different types of studies, including original papers, literature reviews and meta-analyses.

To better understand the link between genetic factors and human disorders and diseases, we studied the correlation among TL, three SNPs that may influence the risk factors for LRDs [9, 16–18] or the onset of LRDs [19], LRD-related physiological and anthropometric

measurements, and the onset of LRDs themselves in a Japanese population from limited areas near our university in one year at a single facility.

## Materials and methods

### Subjects

Since March 2011, an annual university-sponsored health course for community members has been conducted [20]. In each course, which was held ten times per year, health seminars and point-of-care testing were performed. In 2015, a total of 223 attendants were enrolled, and 122 out of 143 attendants in the year (85.3%) agreed to participate in an analysis of salivary TL and SNPs of the *ADIPOQ*, *SIRT1* and *FOXO3A* genes. The inclusion criteria were Japanese descent, clinical stability and sufficient salivary DNA extraction for genetic factor analysis. The exclusion criterion was any acute illness. In two participants, the DNA amount was not enough for analysis, and the remaining 120 subjects were included in the present study (S1 Table). They ranged in age from 41 to 84 years and consisted of 34 men and 86 women with a mean age of 73.3 and 67.8 years, respectively. In three other healthy volunteers, TL was determined by using DNA from saliva and peripheral blood-derived leukocytes.

### DNA extraction from saliva and blood

Participants expectorated at least two ml of passive drool saliva into a sterile, 50 ml polyethylene conical tube (Corning Science, Corning, NY, USA) for five min. These collection tubes were maintained on ice until use. Immediately, one ml of saliva was mixed with an equal volume of nuclei lysis buffer (50 mM Tris-HCl, 50 mM EDTA, 50 mM sucrose, 100 mM NaCl, 2.4% SDS, 550 μg/ml proteinase K, pH 8.0) [21]. After digestion of the cell lysate overnight at 53˚C, 400 μl of 5 M NaCl was added to the mixture, shaken vigorously for 15 s and placed on ice for 30 min. The DNA in the supernatant was extracted by using the GenElute Mammalian Genomic DNA Miniprep Kit (Sigma-Aldrich Co., LLC., Tokyo, Japan). The concentration of the isolated DNA was determined using a NanoDrop Spectrophotometer (Thermo Scientific, Waltham, MA, USA).

Heparinized venous blood was collected from three healthy adult volunteers. Every 4 ml of blood was mixed with 1 ml of 5% (w/v) dextran (MW 266,000, Sigma-Aldrich) in phosphate-buffered saline (PBS) (Gibco, Grand Island, NY, USA), incubated for 30 min at room temperature, and the upper layer containing whole leukocytes was harvested [22]. In one volunteer, the whole leukocyte fraction in PBS was further layered onto Ficoll-Hypaque (Ficoll-Paque Plus, 1.077 g/ml, GE Healthcare Life Sciences, Buckinghamshire, England) and centrifuged at 400×g for 30 min. The whole leukocytes resolved into two distinct bands, the upper containing mononuclear cells (MNCs) and the lower containing neutrophils, and either fraction was harvested. MNCs in PBS with 10% fetal calf serum (Gibco) were further placed in Falcon culture dishes (Becton Dickinson Labware, Oxnard, CA, USA) and incubated for 2 h at 37˚C in 5% $CO_2$. After incubation, the supernatant with the lymphocyte fraction was collected and the nonadherent cells were further removed by vigorous washing with PBS. After the addition of 3 ml fresh warm 0.05% Trypsin-EDTA (Gibco) to the flask, the cells were further incubated for five min at 37˚C in 5% $CO_2$ and the adherent cells (monocyte fraction) were collected by tapping the side of the flask. As in the case of saliva, cellular DNA was extracted, and the concentration of the isolated DNA was determined. After blood and saliva cells bound on a glass slide by Cytospin™ 4 Cytocentrifuge (Thermo Fisher Scientific K.K., Tokyo, Japan), Wright-Giemsa staining (Muto Pure Chemicals Co., Ltd., Tokyo, Japan) was performed, and 100 nucleated cells were counted and classified for each of the three slide glass specimens by using an inverted microscope (Eclipse E600, Nikon, Tokyo, Japan) at 1,000× magnification.

## Telomere length determination by the monochrome multiplex quantitative polymerase chain reaction method

TL quantitative polymerase chain reaction (QPCR) was performed with a Chromo4™ Real-Time PCR Detection System (Bio-Rad, Tokyo, Japan) [23]. The telomere primers (*telg* and *telc*, final concentrations 900 nM each) were used for the telomere signal and a set of single-copy gene (*scg*) primers within *β–globin genes* (final concentrations 200 nM each) were used as a reference with Power SYBR™ Green PCR Master Mix (Applied Biosystems, Foster City, CA, USA).

The PCR cycle for monochrome multiplex QPCR (MMQPCR) was as follows: Stage 1: 1 cycle of 2 min at 95˚C; Stage 2: 2 cycles of 5 s at 94˚C and 15 s at 49˚C; and Stage 3: 40 cycles of 5 s at 94˚C, 10 s at 62˚C, 15 s at 74˚C with signal acquisition (for the amplification of telomere template), 10 s at 84˚C, and 15 s at 88˚C with signal acquisition (for the amplification of the scg template). The same standard genomic DNA was used to establish two standard curve reactions in every plate in the study, one for the telomere signal and one for the *scg* signal, which were used for calculation of the T/S ratios (ratios of "telomere signals per *scg* signals"). As each experimental sample was assayed in triplicate, three T/S results were obtained for each sample; the final reported result for a sample in a given run is the average of the three T/S values and was named relative telomere length (RTL). The average T/S is expected to be proportional to the average TL per cell. Samples with an RTL > 1.0 have an average TL greater than that of the standard DNA; samples with an RTL < 1.0 have an average TL shorter than that of the standard DNA.

## TaqMan® PCR assay

Genotyping for *ADIPOQ rs1501299*, *Sirt-1 rs7895833* and *FOXO3A rs2802292* was performed using Custom TaqMan® SNP Genotyping Assays (Applied Biosystems) in which a fluorogenic probe consisting of an oligonucleotide labeled with both a fluorescent reporter dye (FAM or VIC) and a quencher dye is included in a typical PCR [24]. Amplification of the probe-specific product causes cleavage of the probe, generating an increase in reporter fluorescence [25]. Each primer and probe set was used in the TaqMan® SNP Genotyping Assays (ID: C___7497299_10, C__29163689_10 and C__16097219_10; Applied Biosystems) in accordance with the information on the Applied Biosystems website (http://www.appliedbiosystems.com).

PCR was performed according to the manufacturer's instructions provided by Applied Biosystems. In brief, one to 20 ng of template DNA dissolved in 2.25 μl in each well was loaded into 96-well plates for PCR. The total reaction volume was 5 μl after adding 2.5 μl of TaqMan Universal PCR Master Mix (2×) and 0.25 μl of 10× working stock of SNP genotyping assay buffer (Applied Biosystems).

The PCR thermal cycling was as follows: Stage 1: 1 cycle of 2 min at 50˚C; Stage 2: 1cycle of 10 min at 95˚C for initial denaturing; Stage 3: 40 cycles of 15 s at 92˚C and 1 min at 60˚C. Thermal cycling was performed using a Chromo4™ Real-Time PCR Detection System (Bio-Rad). Each 96-well plate contained unknown genotype samples and three reaction mixtures containing the reagents but no DNA (no-template control). The no-DNA control samples were necessary for Sequence Detection System (SDS) 7700 signal processing, as outlined in the TaqMan Allelic Discrimination Guide. The genotypes were determined visually based on the dye component fluorescence emission data depicted in the X-Y scatter plot of the SDS software.

## Physiological and anthropometric measurements

Body height (cm) was measured using a metal height meter (YS-OS, AS ONE Co., Ltd., Osaka, Japan). Weight (kg), body mass index [BMI (kg/m$^2$)] and percentage of body fat (BF) (%) were measured by a dual-frequency body composition analyzer (DC 430A, Tanita Co., Ltd., Tokyo,

Japan) using bioelectrical impedance analysis. Waist circumference (WC) was measured at the umbilicus to the nearest cm with the participant standing and breathing normally. Body temperature (BT) (˚C) was measured using an electronic thermometer inserted into the armpit for 30 sec (ET-C205S, Terumo Co., Ltd., Tokyo, Japan).

Hypertension (HT) was defined as systolic blood pressure (SBP) ≥140 mm Hg, diastolic blood pressure (DBP) ≥90 mm Hg, or the use of antihypertensive medications [26]. After a more than 5 min rest, the SBP and DBP were measured by a trained physician or nurse using a standard mercury sphygmomanometer (SeaStar, 070108041, Tokyo, Japan) with the participant in the sitting position. Two consecutive blood pressure measurements were taken at 2-min intervals, and the lower one was recorded as the blood pressure.

Measurement of carotid maximum intima-medial thickness (max IMT) (mm) through B-mode ultrasonographic imaging was performed using C3cv (Aloka Medical, Ltd., Tokyo, Japan) with a 6-MHz transducer in the right common carotid artery 1.5 to 3.0 cm proximal to the bifurcation [27].

Measurements of the ankle brachial index (ABI) and brachial-ankle pulse wave velocity (PWV) (cm/sec) were performed as follows. Brachial-ankle arterial blood pressures were simultaneously measured using a noninvasive automatic device (model BP-203RPE-III; Nihon Colin, Tokyo, Japan) after a 5-min rest in the supine position [27]. ABI was defined as the ratio of systolic blood pressure in the ankle and the higher side of the two brachial arteries. The PWV on each side was calculated as the transmission distance divided by the transmission time. The transmission time between the right arm and both ankles was calculated using the waveform. The transmission distance between the right brachium and ankle was automatically calculated according to the height of the patient. PWV was evaluated on the higher side.

Saturation of peripheral oxygen (SpO$_2$) was measured using a pulse oximeter device (ko-001, Kohken Medical Co., Ltd., Tokyo, Japan).

## Questionnaire survey

With the help of our trained staff, self-reported questionnaires were completed during attendance, assessing attendants' personal medical history (PMH) and antihypertension medication use. PMH consisted of HT, stroke, acute myocardial infarction (AMI), chronic kidney disease (CKD) with therapy for edema, hyperkalemia or anemia [28] and cancer (S2 Table). The score for PMH was defined as the sum of the number of affected LRDs in PMH.

## Ethical considerations

This study was approved by the Ethics Committee of Saitama Prefectural University (No. 27503). Before study enrollment, participants and normal volunteers who provided both saliva and peripheral blood samples were asked to sign a consent and assent form that described the background and procedures of the study.

## Statistical analyses

Nonparametric statistics were used. Continuous variables are described as the mean ± standard deviation (SD), and categorical variables are described as proportions. Unadjusted differences between groups were performed by one-way analysis of variance (ANOVA) for continuous variables and chi-square tests for categorical variables. Multivariable linear regression analyses were used to test for adjusted differences with adjustments for the confounding effects of age and sex. Student's *t*-test was performed to compare the differences in means between the two groups. A two-tailed *p* value less than 0.05 was used to determine significance. Statistical analyses were performed by IBM SPSS Statistics version 23.

## Results

### Salivary and blood relative telomere length

In any of the three volunteers studied, there was no significant difference in RTL between the DNA extracted from saliva and blood-derived whole leukocytes (S1A Fig). Similarly, in one volunteer, there was no significant difference in RTL between the DNA extracted from blood-derived neutrophils, lymphocytes and monocytes (S1B Fig).

### Relative telomere length and LRD-related physiological and anthropometric measurements

First, we examined the correlation between RTL and age. Age and RTL were inversely correlated in men ($p = 0.049$, r = − 0.340, y = −9.258x + 82.429) and all participants ($p = 0.034$, r = − 0.192, y = −7.711x + 76.857) but not in women ($p = 0.073$, r = − 0.194, y = −8.235x + 75.766) (Fig 1 and S1 Table).

Next, we examined the association between RTL and LRD-related physiological and anthropometric measurements. In men, the long RTL group (n = 17) had a significantly greater height (167.5 ± 6.6 *vs* 162.1 ± 5.3 cm) and had a significantly lower pulse rate (65.3 ± 12.6 *vs* 73.7 ± 15.5/min) than the short RTL group (n = 17) before ($p = 0.011$) and after ($p = 0.016$) and before ($p = 0.041$) but not after ($p = 0.092$) age adjustment (S3 Table). RTL was not associated with other LRD-related physiological or anthropometric measurements.

### Relative telomere length and score for personal medical history

Frequencies of HT for men, women and all participants according to RTL quartiles are shown in Fig 2A. The frequency of HT decreased in the order of 1st, 2nd, 3rd and 4th RTL quartiles except for 4th RTL quartiles for women and all participants. Longer RTL quartiles had an increasingly lower frequency of HT, especially for men. The *p* values for men, women and all participants after adjustments for both sex and age were 0.130, 0.723 and 0.270, respectively. In men, the frequency of HT was significantly lower in the long RTL group (3rd and 4th RTL quartiles) than in the short group (1st and 2nd RTL quartiles) with unadjusted *p* value of 0.039, and the difference in the frequency of HT between the two groups was of borderline statistical significance after adjustment for age ($p = 0.057$) (Fig 2B). The *p* values for women and all participants after adjustments for both sex and age were 0.597 and 0.146, respectively.

In both men and all participants, the PMH score was also significantly higher in the short RTL group than in the long RTL group after adjustments for both age and sex with respective *p* values of 0.004 and 0.029 (Table 1).

For men, women and all participants, the score for personal medical history (PMH) was shown by relative telomere length (RTL) as shown in the materials and methods. The score for PMH was determined and is expressed as the mean ± standard deviation of the sum of the number of affected lifestyle-related diseases in PMH as shown in the materials and methods. The number of participants whose PMH score was completed was shown in parentheses. The *p* value was computed using multivariable linear regression analysis to investigate the association between PMH score and RTL, and was expressed after adjustments for both sex and age.

### SNPs, LRD-related physiological and anthropometric measurements and scores for personal medical history

For *ADIPOQ rs1501299*, the ABI of men was significantly higher in carriers of the T/T genotype than in those with the G/G and G/T genotypes, with *p* values of 0.001 and 0.000, respectively (Table 2). For *SIRT1 rs7895833*, the BMI and WC of men were significantly higher in

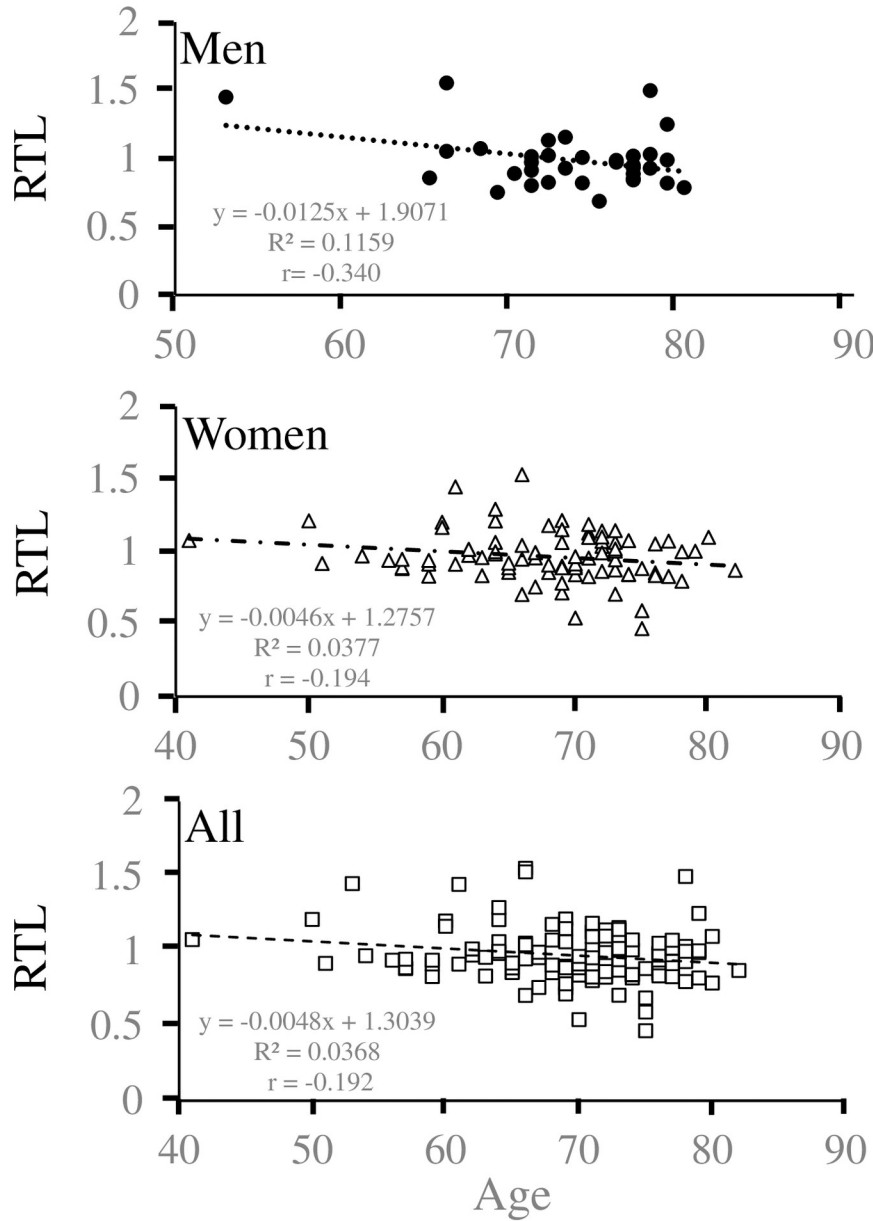

**Fig 1. Cross-sectional analysis of age-dependent shortening of salivary relative telomere length.** For each point, the age (yr) of the participant (x-axis) is plotted against salivary relative telomere length (RTL) (y-axis) for men (upper column), women (middle column), and all participants (lower column). The linear regression equation and correlation coefficient were determined using Microsoft Excel.

carriers of the A/G genotype than in those with the G/G genotype, with respective *p* values of 0.048 and 0.032. The PWV of all participants was also significantly higher in carriers of the A/G genotype than in those with the G/G genotype, with a *p* value of 0.035. For *FOXO3A rs2802292*, the BT of women and $SpO_2$ of all participants were significantly lower in carriers of the G/T genotype than in those with the T/T genotype, with respective *p* values of 0.039 and 0.032. *ADIPOQ rs1501299* in men, but not the other two SNPs, was significantly associated with the PMH score by genotype (*p* = 0.013).

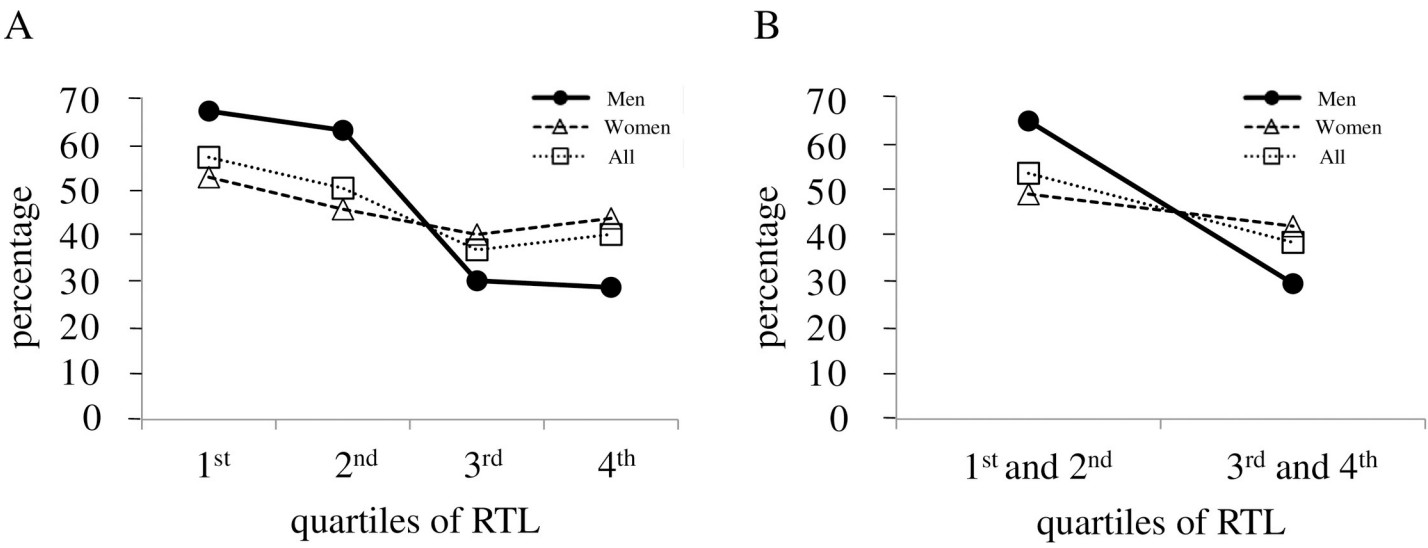

**Fig 2. Frequencies of hypertension by quartiles of salivary relative telomere length. (A)** Frequencies of hypertension (HT) for men, women and all participants in the 1st [66.7% (6/9), 52.4% (11/21) and 56.7% (17/30), respectively], 2nd [62.6% (5/8), 45.5% (10/22) and 50.0% (15/30), respectively], 3rd [30.0% (3/10), 40.0% (8/20) and 36.7% (11/30), respectively] and 4th [28.6% (2/7), 43.5% (10/23) and 40.0% (12/30), respectively] quartiles of relative telomere length (RTL) are shown. The frequency of HT was determined by using data in S1 Table. The RTL levels increase in the order of 1st, 2nd, 3rd and 4th RTL quartiles. For men, women and all participants, the $p$ value was computed using multivariable linear regression analyses to investigate the association between frequency of HT and RTL, and was expressed after adjustments for both sex and age. **(B)** Frequencies of HT for men, women and all participants are also shown in two groups, one with a combination of 1st and 2nd RTL quartiles [64.7% (11/17), 48.8% (21/43) and 53.3% (32/60), respectively], and the other with a combination of 3rd and 4th RTL quartiles [29.4% (5/17), 41.9% (18/43) and 38.3% (23/60), respectively]. The $p$ value was determined and expressed as shown in **(A)**.

In men, women and all participants, the results for LRD-related physiological and anthropometric measurements, RTL and frequency of HT not associated with any of the three SNPs by genotype are shown in S4 Table.

### Reproducibility of salivary relative telomere length measurements

Finally, to examine inter-assay reproducibility, we repeated the measurements of salivary RTL in the same 120 DNA samples as in Fig 1, again in triplicate, on another day. Fig 3 shows the strong correlation between the salivary RTL determined by the first and second runs ($R^2 = 0.9061$).

### Discussion

We successfully determined RTL and genotypes of the three SNPs by using DNA from passive drool saliva in any of the 120 participants. However, caution is warranted when comparing TL measured from saliva obtained through passive drool and saliva obtained using swabs or sponges as the percentage of buccal cells in the latter is significantly higher than that of the former [29]. We confirmed in three volunteers that there was no significant difference in RTL

**Table 1. The association of score for personal medical history with relative telomere length.**

| Sex | Relative telomere length | | p value |
|---|---|---|---|
| | **short** | **long** | |
| Men | 0.941±0.556 (17) | 0.353±0.493 (17) | 0.004 |
| Women | 0.692±0.694 (39) | 0.524±0.632 (41) | 0.33 |
| All | 0.768±0.660 (56) | 0.483±0.599 (58) | 0.029 |

**Table 2. The association of SNPs with LRD-related physiological and anthropometric measurements and scores for personal medical history.**

| SNPs | Sex | Genotype no. (%) n = 120 | Age y.o. n = 120 | BMI n = 110 | WC n = 120 | BT n = 106 | ABI n = 120 | PWV n = 120 | SpO$_2$ n = 119 | PMH score n = 114 |
|---|---|---|---|---|---|---|---|---|---|---|
| ADIPOQ | Men | GG 18 (53) | 75±4 | 23.3±1.5 | 88±5 | 36.1±0.8 | 1.15±0.06[a] | 1656±288 | 96.3±1.4 | 0.89±0.58[c] |
| | | GT 14 (41) | 72±7 | 22.2±2.6 | 83±8 | 36.1±0.6 | 1.14±0.06[b] | 1545±224 | 96.5±1.4 | 0.356±0.50[c] |
| | | TT 2 (6) | 73±3 | 20.6±6.4 | 79±16 | 36.1±0.1 | 1.38±0.20[a, b] | 1472±33 | 96.5±0.7 | 0.50±0.71[c] |
| | Women | GG 56 (65) | 68±7 | 21.8±3.0 | 83±9 | 36.0±0.5 | 1.12±0.07 | 1574±338 | 96.2±1.3 | 0.61±0.64 |
| | | GT 21 (24) | 65±9 | 22.1±3.1 | 85±10 | 36.1±0.5 | 1.11±0.06 | 1471±300 | 96.8±1.0 | 0.45±0.61 |
| | | TT 9 (11) | 70±5 | 21.8±2.2 | 80±8 | 35.7±0.7 | 1.09±0.09 | 1813±357 | 96.4±1.6 | 1.00±0.87 |
| | All | GG 74 (62) | 70±7 | 22.2±2.7 | 84±9 | 36.0±0.6 | 1.13±0.07 | 1594±326 | 96.3±1.3 | 0.68±0.63 |
| | | GT 35 (29) | 68±7 | 22.1±2.9 | 84±10 | 36.1±0.6 | 1.12±0.06 | 1500±271 | 96.7±1.2 | 0.41±0.56 |
| | | TT 11 (9) | 70±5 | 21.5±3.0 | 80±9 | 35.6±0.6 | 1.14±0.15 | 1751±348 | 96.4±1.4 | 0.91±0.83 |
| SIRT1 | Men | AA 3 (9) | 71±4 | 21.0±2.0 | 82±6 | 35.5±2.0 | 1.16±0.08 | 1619±156 | 96.0±1.0 | 0.67±0.58 |
| | | AG 12 (35) | 75±4 | 24.1±2.1[d] | 90±5[e] | 36.2±0.7 | 1.18±0.12 | 1703±201 | 96.1±1.6 | 0.75±0.45 |
| | | GG 19 (56) | 72±6 | 22.0±2.3[d] | 82±8[e] | 36.1±0.4 | 1.15±0.06 | 1531±286 | 96.7±1.3 | 0.58±0.69 |
| | Women | AA 7 (8) | 73±2 | 23.4±1.5 | 85±5 | 35.8±0.8 | 1.11±0.07 | 1590±284 | 96.0±2.2 | 0.86±0.90 |
| | | AG 39 (45) | 68±6 | 21.5±2.7 | 83±9 | 36.1±0.5 | 1.12±0.06 | 1651±355 | 96.4±1.3 | 0.66±0.68 |
| | | GG 40 (47) | 66±8 | 22.0±3.2 | 83±10 | 36.0±0.5 | 1.11±0.07 | 1496±323 | 96.4±1.1 | 0.53±0.60 |
| | All | AA 10 (8) | 73±3 | 22.6±2.0 | 84±5 | 35.7±1.2 | 1.12±0.07 | 1599±244 | 96.0±1.8 | 0.80±0.79 |
| | | AG 51 (43) | 70±7 | 22.2±2.8 | 84±9 | 36.1±0.5 | 1.13±0.08 | 1663±325[f] | 96.3±1.4 | 0.68±0.63 |
| | | GG 59 (49) | 68±8 | 22.0±2.9 | 83±10 | 36.0±0.5 | 1.13±0.08 | 1507±310[f] | 96.5±1.2 | 0.54±0.63 |
| FOXO3A | Men | TT 19 (56) | 73±6 | 22.6±2.3 | 85±8 | 36.0±0.5 | 1.18±0.10 | 1534±165 | 96.6±1.2 | 0.63±0.68 |
| | | GT 15 (44) | 74±4 | 22.7±2.6 | 85±8 | 36.1±0.9 | 1.14±0.06 | 1682±329 | 96.1±1.6 | 0.67±0.49 |
| | Women | TT 50 (58) | 69±7 | 21.9±3.0 | 84±10 | 36.1±0.5[g] | 1.12±0.06 | 1565±336 | 96.6±1.2 | 0.59±0.68 |
| | | GT 36 (42) | 67±7 | 21.9±2.9 | 81±8 | 35.8±0.5[g] | 1.11±0.07 | 1586±350 | 96.1±1.3 | 0.65±0.66 |
| | All | TT 69 (58) | 70±7 | 22.1±2.8 | 84±10 | 36.1±0.5 | 1.14±0.08 | 1556±298 | 96.6±1.2[h] | 0.60±0.67 |
| | | GT 51 (42) | 69±7 | 22.1±2.8 | 83±8 | 35.9±0.7 | 1.12±0.08 | 1614±344 | 96.1±1.4[h] | 0.65±0.61 |

For men, women and all participants, and for any of the three types of SNPs by genotype, data are expressed as the mean ± standard deviation. The *p* value was computed using multivariable linear regression analysis to investigate the association of genotype with lifestyle-related disease-related physiological and anthropometric measurements and scores for personal medical history, and was expressed after adjustments for both sex and age.

Abbreviations: SNPs, single nucleotide polymorphisms; Age, age (yr); RTL, relative telomere length; BMI, body mass index (kg/m$^2$); WC, waist circumference (cm); BT, body temperature (˚C); ABI, ankle brachial index; PWV, brachial-ankle pulse wave velocity (cm/sec); SpO$_2$, saturation of peripheral oxygen (%); PMH, personal medical history.

[a] *p* = 0.001

[b] *p* = 0.000

[c] *p* = 0.013

[d] *p* = 0.048

[e] *p* = 0.032

[f] *p* = 0.03

[g] *p* = 0.039

[h] *p* = 0.032

measured using passive drool saliva and blood-derived whole leukocytes. In one volunteer, the proportion of neutrophils in passive drool saliva was as high as 78.27 ± 6.03% (mean ± SD of triplicate measurements). To date, a growing body of evidence has shown that TL measurement using passive drool saliva instead of blood-derived whole leukocytes is quite useful [30–32].

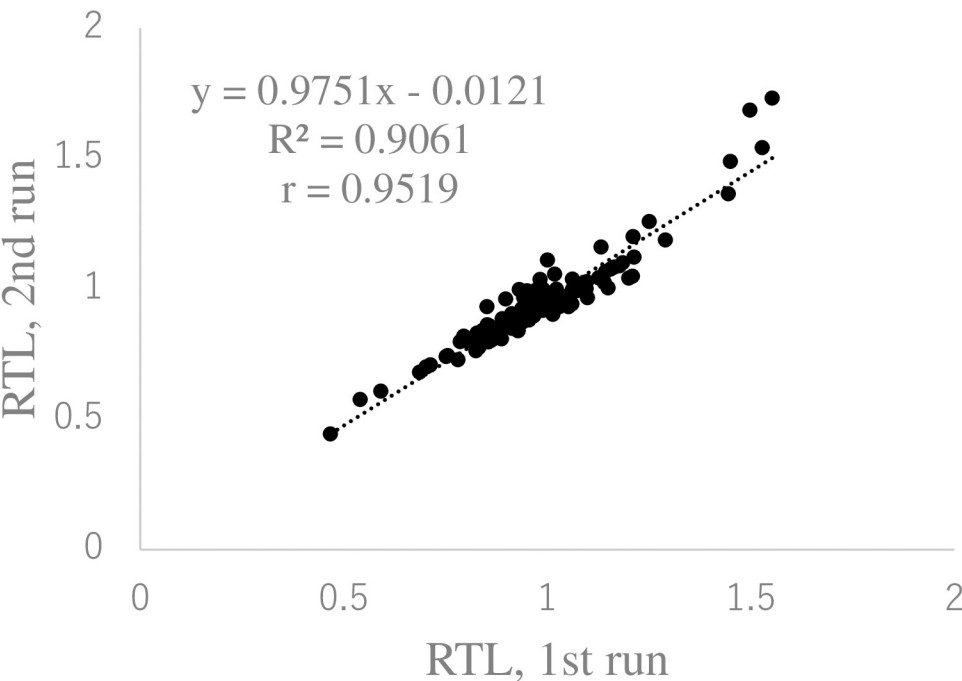

**Fig 3. Reproducibility of salivary relative telomere length in independent runs.** The same 120 salivary DNA samples assayed in Fig 1 were assayed again on different days. The linear regression equation and correlation coefficient were determined using Microsoft Excel.

As previously reported [33], the long RTL group (3rd plus 4th) had a significantly lower frequency of HT than the short group (1st plus 2nd) only in men with unadjusted $p$ value of 0.039, and the difference was of borderline statistical significance after age adjustment ($p = 0.057$). Furthermore, compatible with a previous report [34], our present study found a significant association between the PMH score and RTL in men and all participants. In the present study, LRDs consisted mainly of cardiovascular diseases and cancer. Inconsistent with previous reports [35, 36], however, RTL did not correlate with SBP, DBP or pulse pressure because of the frequent use of antihypertension medication in the present study (40/120; 33.3%).

The overall effect of individually unique environmental factors during adult life, such as energy intake, lifestyle, socioeconomic status and mental stress, is relatively small compared with the joint effect on TL of heritability and shared environmental factors, which is estimated at ~87% [37]. In addition, in recent years, medical treatment and preventive medicine have made remarkable progress [38]. Interestingly, the Mendelian randomization approach, which usually avoids the potential confounding effect of environmental exposure on the relation between phenotypically measured TL and risk of ischemic stroke, has shown that genetically predicted TL is not associated with ischemic stroke [8]. The effect of heritability on TL is presumed to be weaker in the middle-aged to elderly subjects in the present study than in the younger subjects. For these reasons, in the present study as well, it seems likely that RTL appears to be more strongly associated with the combination of heritability and shared environmental factors than heritability or individual-specific environmental factors alone, in terms of predisposition to the development of LRDs.

In the present study, men in the long RTL group were shown to be significantly taller than those in the short RTL group. Although it is undeniable that bone mineral density may be maintained with long TL [39], it would be more correct to say that height is not a risk factor for LRDs.

Associations between the SNPs and LRD-related physiological or anthropometric measurements in the present study were as follows: for *ADIPOQ rs1501299*, in accordance with previous papers [16, 19], men's ABI was higher in the T/T genotype than in the G/G or G/T genotype; for *SIRT1 rs7895833*, consistent with previous papers [17, 18], men's BMI and waist circumference and all participants' PWV were higher in the A/G genotype than in the G/G genotype; for *FOXO3A rs2802292*, women's BT and all participants' SpO$_2$ were lower in the G/T genotype than in the T/T genotype. We confirmed again in the persent study that the Japanese allele frequencies for *SIRT1 rs7895833* [17], which are different from those of Caucasians [18], might explain why Japanese individuals show less marked obesity than Caucasians. Consistent with a role for *FOXO3* in suppression of insulin/insulin-like growth factor and thus growth to extend lifespan, the longevity-associated G allele variant of *FOXO3A rs2802292* seems to be associated with lower BT and SpO$_2$ [9, 40].

Compatible with previous reports [16, 19], only *ADIPOQ rs1501299*, but not the other two types of SNPs showed a significant correlation with both risk factors for and PMH score of LRDs. In any of the three SNPs, there was no significant difference in the frequency of HT or RTL by genotype.

Many studies have shown that TL is a heritable trait, and SNPs in several candidate genes including *TERT* (telomerase reverse transcriptase), *TERC* (telomerase RNA component), *OBFC1* (oligonucleotide/oligosaccharide-binding folds containing one), *CTC1* (conserved telomere maintenance component 1) and *ZNF676* (zinc fingerprotein676) have been identified to be significantly associated with TL [2, 41]. These genes have limited genetic variations and encode proteins that are thought to be involved either in TL maintenance or with telomere binding proteins necessary for telomere stability and structure. Exceptionally, *fat mass- and obesity-associated* (*FTO*) *genes* have been shown to be associated with obesity and TL [7], and the *FOXO3 G* allele of SNP *rs2802292* significantly protected against aged-related TL relative to that of carriers of the common TT genotype [42]. However, we could not find any association between RTL and genotype in any of the three SNPs examined in the present study.

SNPs are the most common type of human genetic variation and have been associated with disease development and phenotypic forecasting [43]. Although GWAS identified genetic variants involved in complex phenotypes, the fraction of heritability of common traits and diseases explained by the identified loci is small [44]. The reason for this small proportion of the heritability of these complex traits is still unclear, but the causes can involve epistatic effects, genetic interactions inside undiscovered pathway or underestimated genotype-environment interactions [45, 46].

*SIRT1* also regulates a stress-response transcription factor, *FOXO3*, thereby modulating cellular senescence/aging, skeletal muscle function, cardiovascular homeostasis, and human longevity [9, 40, 47]. Furthermore, telomeres have been shown to be closely associated with *sirtuin* [48] and *FOXO3* [42]. Therefore, further studies are required to clarify the exact correlation between the two SNPs and TL.

DNA methylation is capable of controlling the gene expression of common traits and influencing the development of aberrant health outcomes under external exposure [49]. Recently, an epigenetic 'mortality risk score' based on whole blood DNA methylation at 10 mortality-related CpG sites has been shown to be strongly associated with TL and even more strongly associated with all-cause mortality [50]. However, TL failed to correlate with all-cause mortality. DNA methylation studies of the SNPs may further clarify the relationship among SNPs, TL and LRDs [51].

In the present study, the frequency of the significant association of genetic factors with LRDs was higher in men than in women. It is presumed that the reason is, in part, that men have a higher prevalence of LRDs, especially cardiovascular disease, and a shorter life

expectancy than women in specific age groups, 50–70 [52]. The number of subjects in this study was small, 34 men and 86 women, with a total of 120, so the results of the study may be limited; however, none of the 34 men were excluded (S1 Table). In any case, further studies are required to confirm our present results.

## Conclusion

Over the course of a year and in a single facility, we showed that RTL and the three types of SNPs determined using saliva are differentially associated with onset of and measured risk factors for LRDs consisting mainly of cardiovascular diseases and cancer in Japanese individuals living in a small area near our university.

## Supporting information

**S1 Fig. Salivary and blood relative telomere length.**
(DOCX)

**S1 Table. Participants' characteristics, relative telomere length and personal medical histories.**
(DOCX)

**S2 Table. Request for personal medical history.**
(DOCX)

**S3 Table. Association between relative telomere length and LRD-related physiological and anthropometric measurements.**
(DOCX)

**S4 Table. LRD-related physiological and anthropometric measurements, relative telomere length and frequency of hypertension not associated with any of the three SNPs by genotype.**
(DOCX)

## Acknowledgments

We thank our lab members for supporting the annual university-sponsored course for community members. We also thank Dr. Yukio Kamezawa (Saitama Prefectural University) for generously providing cell-separation reagents and for helpful discussions on this study.

## Author Contributions

**Conceptualization:** Ken-ichi Tanaka, Hiroyuki Hayashi, Kyoko Arakawa, Ikuo Murohashi.

**Data curation:** Xiao Han, Nobue Sakai.

**Formal analysis:** Ryo Kubota.

**Funding acquisition:** Xiao Han, Ikuo Murohashi.

**Methodology:** Noriko Kawaguchi-Ihara, Ikuo Murohashi.

**Project administration:** Ikuo Murohashi.

**Resources:** Miyuki Seki, Nobue Sakai, Noriko Kawaguchi-Ihara, Kyoko Arakawa.

**Software:** Xiao Han.

**Writing – original draft:** Xiao Han, Ikuo Murohashi.

**Writing – review & editing:** Ikuo Murohashi.

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
