## [Decision Letter · Decision Letter 0]

13 Sep 2020

PONE-D-20-22106

The correlation of salivary telomere length and single nucleotide polymorphisms of

ADIPOQ ,SIRT1 and FOXO3A genes with lifestyle-related diseases in Japanese

PLOS ONE

Dear Dr. Murohashi,

Thank you for submitting your manuscript to PLOS ONE. After careful consideration, we feel that it has merit but does not fully meet PLOS ONE’s publication criteria as it currently stands. Therefore, we invite you to submit a revised version of the manuscript that addresses the points raised during the review process.

We look forward to receiving your revised manuscript.

Kind regards,

Hoh Boon-Peng, PhD

Academic Editor

PLOS ONE

Additional Editor Comments:

1. Authors should revise the language of the manuscript extensively, and send for language editing and proofreading before resubmitting the manuscript.

2. Authors should justify if 120 samples are appropriate to provide sufficient power of study,

3. Comments by the Reviewer#2 on the telomere length estimate should be clarified. In particular, the correlation between the telomere length and disease outcome should be relooked.

Journal Requirements:

Reviewers' comments:

Reviewer's Responses to Questions

**Comments to the Author**

1. Is the manuscript technically sound, and do the data support the conclusions?

Reviewer #1: Yes

Reviewer #2: No

2. Has the statistical analysis been performed appropriately and rigorously? 

Reviewer #1: Yes

Reviewer #2: No

3. Have the authors made all data underlying the findings in their manuscript fully available?

Reviewer #1: Yes

Reviewer #2: No

4. Is the manuscript presented in an intelligible fashion and written in standard English?

Reviewer #1: No

Reviewer #2: No

5. Review Comments to the Author

Reviewer #1: This is an interesting and well conducted study on the correlation of genetic factors (3SNPs) and lifestyle-related diseases in a Japanese cohort. Several physiological parameters have been measured for that. Language is in general acceptable but has to be corrected by a native speaker since articles are often missing or misplaced and there are other language and grammar issues as well.

Although it is very good that you confirmed the correspondence of TL in saliva and blood cells I would suggest to move figure 1 to the supplementary since it is just a methodological detail and only very few samples ( 3 or 1) have been included for blood related markers.

For table 1 please include n-numbers for each group and better describe in the text how exactly you determined the prevalence for LRDs.

Importantly, since your study cohort included much more women than men: is it possible that several significances in men are due to the much smaller sample size?

For table 2 please make the letters the p-values at the end refer to larger since it is very difficult to spot them, perhaps use bold font as well. When you discuss heritability of TL, please make sure to emphasise that the % of heritability decreases with age-it is higher in young people than older and you have a middle-age/older population in your study.

Presentation of figure 3 needs to be improved. For 3A, what is % if you want to present the prevalence? Please clarify! 3B needs its own axis labelling! For both graphs please show error bars and statistical significances.

Reviewer #2: 1) The abstract requires profound revision, dividing sections of introductions, methods, results and conclusions. In this form, the rationale of the study is not reported also.

2) Tables: authors should fix the tables that are very not easy-to-get in these forms.

3) Sample size analysis is lacking in statistical sections. This is relevant for such type of genetic data (and numbers are not so convincing).

4) Major issue: authors aims to address a correlation between a modifiable marker (telomere length) with modifiable exposure to environmental factors, “blocking” for a genetic trait. However, there is a lack of causality which is given by the non-causal effect of polymorphisms. How often are expected these variants to be frequent in your populations? Again (sample size is lacking)? Are the variants in accordance with Hardy-Weinberg equilibrium?

Based on these lacks conclusions in the abstact as well as in the text are not supported by data. Telomeres cannot be defined “independently associated”. Please, I strongly suggest to revise this point.

5) How much authors are confident on the third number decimal as for telomere length? Since data are coming from an “estimated” fold-change of a pcr-based method, generally it would be hard to be so precise. How many triplicates intra- and inter-day have been repeated for telomere length ?

6) The heading of table 1 is not clear: it is reporting a 2X2 association between gender and telomere length, so how do authors comment on the heading of the table?

6. PLOS authors have the option to publish the peer review history of their article (what does this mean?). If published, this will include your full peer review and any attached files.

Reviewer #1: **Yes: **Gabriele Saretzki

Reviewer #2: No

---

## [Author Response · Author response to Decision Letter 0]

24 Oct 2020

‘Response to Reviewers’

Hoh Boon-Peng, PhD

Academic Editor

PLOS ONE, and

Two reviewers

Thank you for your kind and accurate guidance. Responses to academic editor and reviewers are as follows.

Hoh Boon-Peng, PhD

Academic Editor

PLOS ONE

First of all, we apologize for making two mistakes and correcting them. The first is the mistake for statistical processing of quantification of personal medical history. Then, we showed all 120 subjects’ data including age, sex, relative telomere length (RTL) and personal medical history in S1 Table. Due to the correction, the contents of Table 1 have been slightly changed. The second is that we noticed the significant correlation between SNPs ADIPOQ rs1501299 and personal medical history in men. Due to the correction, the contents of Table 2 have been slightly changed. 

Next, we will answer your questions in order below. 

1. We asked American Journal Expert (AJE) for language editing and proofreading. Certificate of the completion of calibration by AJE is attached.

2. We confirmed by using G*Power software (HHU) that 120 samples are appropriate to provide sufficient power of study. In addition, this study was performed in a single institute in one year in Japanese population. This is described in the revised manuscript. 

3. In our paper, a total of 120 subjects were analyzed, 34 men and 86 women. The characteristics of the 120 participants were shown in S1 Table. The description of relative telomere length, three SNPs, hypertension was completed in all 120 subjects. The personal medical history was complete in all 34 men and 80 women, a total of 114. The total of five diseases checked by our trained staff during attendance consisted of hypertension, acute myocardial infarction, stroke and chronic kidney disease, which mainly develop based on vascular lesions, and the rest is cancer (any malignant neoplasms). The sum of the number of affected diseases was compared with the relative telomere length. 

Answer to Journal requirements

1. We corrected manuscript to meet PLOS ONE’s style requirements, as far as possible.

2. Our manuscript has been corrected by American Journal Experts, and the certificate of the completion of calibration is attached.

3. “Request for personal medical history” in both Japanese and English was added and shown in S2 Table.

4. I will soon ensure ORCiD ID.

5. We deleted all “data not shown”. All relevant data were shown in S1, S3 and S4 Tables. 

Answers to the reviewers’ comments.

1. We have rewritten our revised manuscript extensively according to your guidance.

2. In the Materials and Methods, we rewritted statistical analyses properly.

3. All data were added and shown in S1, S3 and S4 tables.

4. We asked American Journal Expert (AJE) for language editing and proofreading. Certificate of the completion of calibration by AJE is attached. In addition, we read the manuscript many times and corrected the mistakes.

5. Answers to review comments.

Reviewer #1

 Thank you for your important advice. We think that our manuscript has been greatly improved owing to the proofreading by American Journal Experts. The certificate of the completion of calibration is attached. 

 Fig 1 has been changed to S1 Fig, as instructed. 

 For Table 1, the number of subjects is shown for each group, as instructed.

 The number of 34 male subjects is not large, but none were excluded. In addition, relative telomere length determination and personal medical history hearing have been completed for all, as shown in S1 Table. We discussed in depth in the Discussion why the frequency of significant differences are higher in men in the present study. 

 According to your suggestion for Table 2, we changed the letters the p-values at the end refer to bold font

As suggested, “The effect of heritability on TL is presumed to be weaker in the middle-aged to elderly subjects in the present study than in the younger subjects.“ has been added to the Discussion.

We are very sorry for making an inappropriate description in Fig 3. We replaced prevalence with frequency (percentage). We added axis labelling to both A and B. In the frequency [for example 66.7% (6/9)] only one number comes out, and it is impossible to show as mean ± SD. Multivariable linear regression analyses were used to identify statistical significance. As the number of statistical comparisons is too large to describe, only one (p = 0.057) is shown. 

Reviewer #2

1) The abstract has been significantly revised, and was divided to introduction, objectives, methods, results and conclusions.

2) Tables 1 and 2 have been improved to the maximum to make them easier to read.

3) We confirmed by using G*Power software (HHU) that 120 samples are appropriate to provide sufficient power of study. In addition, this study was performed at a single facility in one year in a Japanese population. This is described in the revised manuscript. In our paper, a total of 120 subjects were analyzed, 34 men and 86 women. The characteristics of the 120 participants were shown in S1 Table. The description of relative telomere length, three SNPs, hypertension was completed in all 120 subjects. The personal medical history was complete in all 34 men and 80 women, a total of 114. The number of subjects was indicated in Tables 1 and 2, and Tables S3 and S4. We added the description of “The number of subjects in this study was small, 34 men and 86 women, with a total of 120, so the results of the study may be limited.” to the Discussion. 

4) We are very grateful for pointing out a very important issue. We deeply apologize for making an inappropriate statement of “independently associated”. This statement in both Abstract and Discussion has been deleted.

5) We measured RTL according to the method by Cawthon (Nucleic Acids Res 2009; 37:e21, reference 23. In our paper). He showed that the correlation of RTL with terminal restriction fragment length measured by Southern blot was strong (R2 = 0.844). In some samples, we repeated the measurements of RTL, again in triplicate, on another day. For analysis, however, the results of the first measurement were used for any of the subjects.

6) In our present study, in men, women and men and women in total, clinical characteristics of the study subjects including score for personal medical history were compared across the median or quartiles of RTL by using the methods shown in statistical analyses. Then, we consider that the heading of the Table 1 of “The association of score for personal medical history with relative telomere length” is correct. 

6. Do you want your identity to be public for this peer?

Yes

---

## [Decision Letter · Decision Letter 1]

11 Nov 2020

PONE-D-20-22106R1

The correlation of salivary telomere length and single nucleotide polymorphisms of the

ADIPOQ ,SIRT1 and FOXO3A genes with lifestyle-related diseases in a Japanese population

PLOS ONE

Dear Dr. Murohashi,

Thank you for submitting your manuscript to PLOS ONE. After careful consideration, we feel that it has merit but does not fully meet PLOS ONE’s publication criteria as it currently stands. Therefore, we invite you to submit a revised version of the manuscript that addresses the points raised during the review process.

The authors addressed the majority of the comments, however a proper description and interpretation of the results is missing. Most of the results consist of the 2 figures and tables, without a a clear explanation of those rather complex data. This has to be improved and amended to make it easier for the reader to follow the content of the study which is pretty complex.

We look forward to receiving your revised manuscript.

Kind regards,

Hoh Boon-Peng, PhD

Academic Editor

PLOS ONE

Additional Editor Comments (if provided):

The authors addressed the majority of the comments, however a proper description and interpretation of the results is missing. Most of the results consist of the 2 figures and tables, without a a clear explanation of those rather complex data. This has to be improved and amended to make it easier for the reader to follow the content of the study which is pretty complex.

Reviewers' comments:

Reviewer's Responses to Questions

**Comments to the Author**

1. If the authors have adequately addressed your comments raised in a previous round of review and you feel that this manuscript is now acceptable for publication, you may indicate that here to bypass the “Comments to the Author” section, enter your conflict of interest statement in the “Confidential to Editor” section, and submit your "Accept" recommendation.

Reviewer #1: (No Response)

2. Is the manuscript technically sound, and do the data support the conclusions?

Reviewer #1: Yes

3. Has the statistical analysis been performed appropriately and rigorously? 

Reviewer #1: I Don't Know

4. Have the authors made all data underlying the findings in their manuscript fully available?

Reviewer #1: Yes

5. Is the manuscript presented in an intelligible fashion and written in standard English?

Reviewer #1: Yes

6. Review Comments to the Author

Reviewer #1: Although the authors addressed the majority of my initial comments, I find that a proper description and interpretation of the results is really missing. Most of the results consist of the 2 figures and tables, but I a missing an explanation of those rather complex data. For example, for fig. 2 the authors state that they have performed various complex statistical methods while just giving 1 non-significant p-value (a, p=0.057). This has to be much better described and understandably interpreted for the reader, Normally, the results text mentions every single figure and table while here the authors write something in the text which I find difficult to relate to particular graphs, tables or statistical comparisons. Please try to give your results in better way as it is now.

Also, the heading for fig. 2 reads as "hypertensive frequency" which is my vie should be rather "Frequencies of hypertension" since the frequency isn't really suffering from hypertension.

7. PLOS authors have the option to publish the peer review history of their article (what does this mean?). If published, this will include your full peer review and any attached files.

Reviewer #1: **Yes: **Gabriele Saretzki

---

## [Author Response · Author response to Decision Letter 1]

24 Nov 2020

We believe that we have responded properly to any of the questions and requests, and now the submitting paper has been markedly improved owing to the help by academic editor and rewieers.

---

## [Decision Letter · Decision Letter 2]

26 Nov 2020

The correlation of salivary telomere length and single nucleotide polymorphisms of the

ADIPOQ ,SIRT1 and FOXO3A genes with lifestyle-related diseases in a Japanese population

PONE-D-20-22106R2

Dear Dr. Murohashi,

We’re pleased to inform you that your manuscript has been judged scientifically suitable for publication and will be formally accepted for publication once it meets all outstanding technical requirements.

Kind regards,

Hoh Boon-Peng, PhD

Academic Editor

PLOS ONE

Additional Editor Comments (optional):

Reviewers' comments:

Reviewer's Responses to Questions

**Comments to the Author**

1. If the authors have adequately addressed your comments raised in a previous round of review and you feel that this manuscript is now acceptable for publication, you may indicate that here to bypass the “Comments to the Author” section, enter your conflict of interest statement in the “Confidential to Editor” section, and submit your "Accept" recommendation.

Reviewer #1: (No Response)

2. Is the manuscript technically sound, and do the data support the conclusions?

Reviewer #1: (No Response)

3. Has the statistical analysis been performed appropriately and rigorously? 

Reviewer #1: (No Response)

4. Have the authors made all data underlying the findings in their manuscript fully available?

Reviewer #1: (No Response)

5. Is the manuscript presented in an intelligible fashion and written in standard English?

Reviewer #1: (No Response)

6. Review Comments to the Author

Reviewer #1: (No Response)

7. PLOS authors have the option to publish the peer review history of their article (what does this mean?). If published, this will include your full peer review and any attached files.

Reviewer #1: **Yes: **Dr. Gabriele Saretzki

---

## [Editor Report · Acceptance letter]

11 Jan 2021

PONE-D-20-22106R2 

The correlation of salivary telomere length and single nucleotide polymorphisms of the *ADIPOQ, SIRT1* and *FOXO3A* genes with lifestyle-related diseases in a Japanese population 

Dear Dr. Murohashi:

I'm pleased to inform you that your manuscript has been deemed suitable for publication in PLOS ONE. Congratulations! Your manuscript is now with our production department. 

Kind regards, 

on behalf of

Dr. Hoh Boon-Peng 

Academic Editor

PLOS ONE